# Bridging the Perception Gap: Probe-Guided Data Optimization Framework for Robotic Imitation Learning

## Abstract

Imitation learning allows robots to acquire complex manipulation skills from human demonstrations. However, traditional data collection methods often haven't account for the "perceptual gap" between humans and robots, which leads to models that don't perform as expected. To solve this, we introduce Policy-Intent Probe (PIP). This method trains a proxy model with a small amount of demonstration data, then quantifies the model's perceptual capabilities based on its policy distribution. Based on the model's feedback, we divide the task's operational space into a Robust Operation Zone (ROZ) and a Non-Robust Operation Zone (NROZ). By standardizing the trajectories in the NROZ and waiting until the robot enters the ROZ to perform precise operations, we have optimized the data collection trajectories. Furthermore, aided by PIP, we can correctly carry out subtask segmentation. We can supplement data collection based on subtask complexity, which enhances the model's generalization and robustness. By cleaning only the subtask data containing anomalous trajectories or failure, we minimize data loss. Based on an empirical evaluation on three real-world tasks, we proved that perceptual capabilities can affect a task's success rate and that arbitrary subtask decomposition can lead to negative consequences. Our model-in-the-loop data optimization framework can significantly boost the success rate of long-horizon precision manipulation tasks, enhance model robustness, and increase data collection efficiency.

## 1 Introduction

Robotic imitation learning(IL) Hussein et al. (2017); Osa et al. (2018); Chi et al. (2023); Xue et al. (2025) has emerged as a prevalent approach for solving complex manipulation tasks that are difficult to model. A key method within this field is behavioral cloning(BC) Pomerleau (1988) , which allows us to learn a state-to-action policy directly from demonstration data collected from human experts. While advanced VLA models like RT-2 Zitkovich et al. (2023), $\pi_0$ Black et al. (2024), Helix Figure (2024), and GR00T N1 Bjorck et al. (2025) have emerged, with parameters now rivaling those of smaller large language models (LLMs), they have yet to demonstrate the "emergent intelligence" seen in their LLM counterparts. This is primarily due to two factors: the limited volume of available data and the absence of high-quality demonstration data.

Methods that use simulation platforms like RoboTwin Mu et al. (2025); Chen et al. (2025) and RoboGSim Li et al. (2024) can generate large amounts of demonstration data. However, it's difficult to accurately model complex tasks in the real world, and the sim-to-real gap often leads to a performance drop when these models are deployed on real-world tasks. The availability of open-source projects like Gello Wu et al. (2024b), UMI Chi et al. (2024), and Mobile ALOHA Fu et al. (2024) has made it more convenient to collect real-world data. However, current data collection paradigms have inherent flaws: we fail to account for the robot's actual perceptual capabilities during the collection process, and data is gathered on an entire-task basis. This leads to poor performance when models are deployed in the real world. To bridge this performance gap, we are forced into a vicious cycle of continuously collecting more data. The perceptual gap Teichmann et al. (2021) between humans and robots is a key, often overlooked, factor that significantly impacts a model's task success rate. This gap can cause the model to fail to output the correct action sequence as expected during real-world deployment, which in turn affects its reliability in complex tasks.

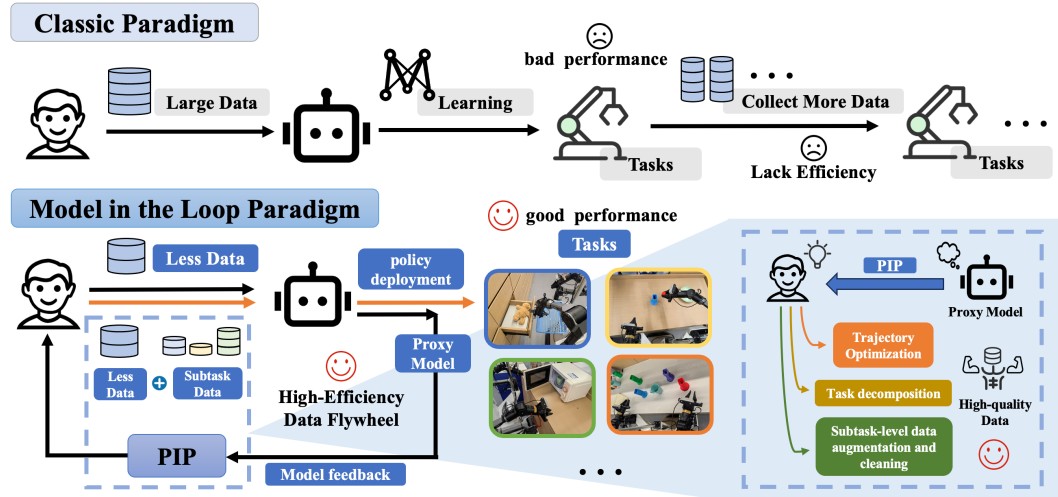

Figure 1: **Probe-guided data optimization framework overview.** Traditional data collection is decoupled from model training. This approach is inefficient and hinders performance improvement. Our optimization framework uses a proxy model to gain insights into the model's perceptual capabilities. This feedback is then used to optimize our data collection process, forming a data flywheel that significantly boosts both data collection efficiency and overall model performance.

This paper presents a Probe-Guided Data Optimization Framework. The framework uses the policy output of a proxy model as feedback to optimize the data collection process. This approach helps bridge the perceptual gap between humans and robots, significantly improving data quality and data collection efficiency. Ultimately, this forms an efficient data flywheel Luo et al. (2024); Wang et al. (2024), and significantly boosts task success rate. Our main methods are summarized as follows:

- **Acquiring Perceptual Capabilities:** We first train a proxy model Wu et al. (2023); Tan et al. (2024b) (which shares the same architecture as the deployed model) using some complete episodes from tasks. Building on this, we use our proposed Policy Intent Probe (PIP) to gain deep insights into the model's perceptual capabilities. PIP, which is based on the Kernel Density Estimation (KDE) Kim & Scott (2012); Chen (2017) method, analyzes the proxy model's policy action distribution given a real demonstration state. This allows us to determine whether the model can execute the correct action based on the current state.

- **PIP-Guided Trajectory Optimization:** Based on our proposed PIP analysis, we divide the task workspace into Robust Operation Zones (ROZ) and Non-Robust Operation Zones (NROZ). The ROZ represents regions where the model can execute correct actions with high confidence, while the NROZ indicates areas of unstable model behavior. By standardizing trajectories within the non-robust operation zones and ensuring critical precise operations are performed in the robust operation zones, we effectively optimize the entire data collection process and bridge the perceptual gap between humans and robots.

- **PIP-Aided Subtask Decomposition:** To further improve data collection efficiency and model robustness, we perform subtask decomposition at key moments where the state space Hamilton (1994) is relatively small and located within the ROZ. This perception-based segmentation method not only ensures smooth transitions and fluid actions between subtasks but also fundamentally prevents the motion freezes and task failures that are often caused by blind segmentation.

- **Subtask-Level Data Optimization:** 1) Subtask-level Data Augmentation: Based on subtask difficulty, we strategically collect data and perform state supplementation Yang et al. (2025) at locations that require precise operations. This improves data collection efficiency and enhances the model's robustness. 2) Subtask-level Data Cleaning: We use PIP analysis to precisely determine which data should be cleaned. We only remove subtask data containing failed or anomalous trajectories, rather than entire task data, which significantly increases data utilization.

## 2 RELATED WORKS

Data collection is inherently a resource-intensive and challenging endeavor. Especially for long-horizon tasks, this not only requires a vast amount of demonstration data, but also demands that the operation be highly precise and coherent. Teleoperation allows experts to directly control real robots in order to collect high-quality demonstrationsZhao et al. (2023a); Wu et al. (2024a); Lin et al. (2024); Qin et al. (2023); Du et al. (2022); George et al. (2025). The data obtained via teleoperation faithfully reflects various dynamics and interactions. However, expert cost is high, data collection rate is low, and special hardware or interfaces are required—making large-scale deployment prohibitively expensive. Because collecting demonstration data in the real world is very costly, many studies have adopted simulation instead. Sim-to-Real methods train policies or generate demonstration data in a simulated environmentMandlekar et al. (2023b; 2021); Jiang et al. (2025); Garrett et al. (2024); Ho et al. (2021); Jia et al. (2022), randomizing appearance, lighting, and physical parameters in the simulatorNasiriany et al. (2024); Tan et al. (2024a), and then apply them in real-world settings. Such methods can generate large datasets cheaply and quickly, and are widely used in tasks such as visual localization and graspingJiang et al. (2024); Dalal et al. (2023); Du et al. (2023). However, the reality gap in dynamics, sensors, lighting is hard to bridgeHo et al. (2021); Aljalbout et al. (2024). Policies trained in simulation often suffer severe failures when deployed in real environments. Human-in-the-Loop methods incorporate human feedback during learning or execution Mandlekar et al. (2020); Wu et al. (2025), such as interventionsCui et al. (2023); Mandlekar et al. (2023a), correctionsChisari et al. (2022); Liu et al. (2023); Hoque et al. (2023), or preference feedbackLiu et al. (2022; 2024); Jiang et al. (2024), not only offline demonstrations. Such methods significantly reduce annotation cost and improve data-efficiency. However, the approach remains inherently contingent on continuous human supervision and monitoring. Issues such as feedback latency, significant quality variations in human corrections can adversely affect the performance and consistency of the learned policy. The method proposed in this paper is a model-in-the-loop data collection approach that leverages a model's actual perceptual capabilities to optimize data collection trajectories, thereby significantly boosting task success rate and data collection efficiency.

## 3 METHODOLOGY

In this section, we will introduce our method for acquiring a model's perceptual capabilities and clarify why neglecting this ability impacts a model's performance during real-world deployment. Subsequently, we leverage the model's perceptual capabilities to optimize data collection trajectories. In addition, we perform a reasonable subtask decomposition to enable efficient data augmentation and data cleaning at the subtask-level. The entire process optimizes data collection through feedback from a proxy model, which forms our Model-in-the-Loop Data Optimization Framework.

### 3.1 POLICY INTENT PROBE (PIP)

The paradigm of imitation learning for robotics begins with an offline dataset of expert demonstrations, $\mathcal{D} = \{(s_t, L, A_t)\}$. Each sample in this dataset contains a multimodal state observation $s_t = (I_t, q_t, x_t)$, which encompasses visual, proprioceptive, and other modalities; a language instruction $L$; and a future $N$-step sequence of expert actions, referred to as an **action chunk** Zhao et al. (2023b), $A_t = (a_t, \ldots, a_{t+N})$.

The objective of imitation learning is to optimize a parameterized policy, $\pi_\theta$, to reproduce the expert's behavior. Theoretically, this is achieved by minimizing the negative log-likelihood (NLL) of the expert actions on the dataset $\mathcal{D}$:

$$\mathcal{L}_{\text{NLL}}(\theta) = -\mathbb{E}_{(s,L,A)\sim\mathcal{D}}\left[\log p_\theta(A|s, L)\right] \tag{1}$$

In practice, modern VLA models, such as those based on **Diffusion** or **Flow Matching**, approximate this objective by optimizing a more tractable surrogate loss. These models learn a vector field $v_\theta$ and minimize the discrepancy between it and a target vector field $u_\tau$:

$$\mathcal{L}_{\text{generative}}(\theta) = \mathbb{E}\left[\|v_\theta(A(\tau), \tau, s, L) - u_\tau\|^2\right] \tag{2}$$

By optimizing this surrogate loss, the policy implicitly learns the complex conditional probability distribution $p_\theta(A|s, L)$ underlying the expert data. During inference, the policy generates an action

chunk from this implicit distribution via a process of **stochastic sampling**: it starts with a random vector sampled from a simple prior (e.g., a Gaussian distribution) and transforms this noise into a final action sequence by solving the learned Ordinary Differential Equation(ODE).

Although imitation learning helps robots perform tasks, a gap often appears between their behavior and human expectations in practice, leading to poor performance. This problem stems from a perceptual gap between robots and humans. For this reason, we urgently need a way to decode the robot's decision-making process.

We propose the **Policy-Intent Probe (PIP)**, a method designed to analyze and decode the complex, **multimodal action distributions** Chi et al. (2023) of a policy, $p_\theta(A|s, L)$. The core workflow of this method consists of three steps:

1. **Action Chunk Sampling**: For a given state observation $s_t$ and language instruction $L$, we perform $N$ stochastic forward passes through the policy $p_\theta(A|s_t, L)$ to generate a set of $N$ action chunk samples, $\{A_1, A_2, \ldots, A_N\}$.

2. **Key Action Extraction & Dimensionality Reduction**: Given that the full action chunks $A_i$ are too high-dimensional for direct analysis, we first select a representative waypoint from each chunk (e.g., the action at the midpoint of the prediction horizon, $a_{t+H/2}$) as the object of our analysis. As this action is still a high-dimensional vector (e.g., 7-DoF or 14-DoF), we employ **Principal Component Analysis (PCA)** Abdi & Williams (2010) to project the $N$ action vectors into a 2D embedding space, capturing their principal variance.

3. **Kernel Density Estimation & Analysis**: Within this 2D embedding space, we apply **Kernel Density Estimation (KDE)** for analysis. We visualize the resulting probability density as a 2D heatmap and contour plot to reveal the policy's overall modal structure, uncertainty, and policy preferences. To provide a clearer view of the distribution along the primary axis of variation, we also plot the 1D KDE curve along the first principal component (PC1).

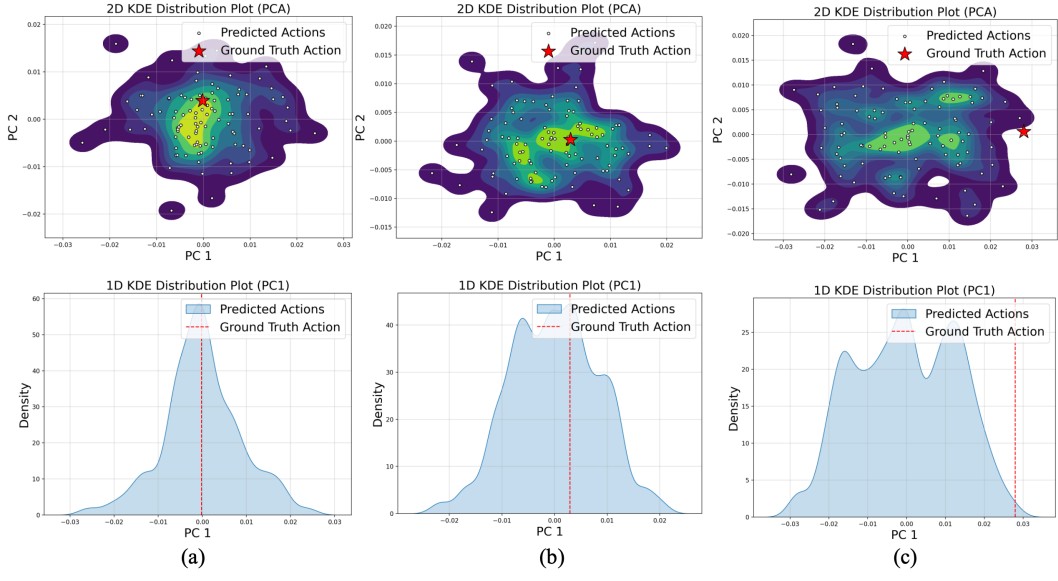

(a)          (b)          (c)

Figure 2: **Commmon KDE Result Plots** (a) shows a highly concentrated multimodal action distribution where the model's predicted actions are very close to the ground truth, indicating an effective action policy. (b) the distribution is slightly dispersed but the predictions remain accurate, which demonstrates a correct policy with multiple feasible action choices. (c) the distribution is highly dispersed and the predictions deviate significantly from the ground truth, indicating the model's inability to produce a correct policy based on the current state.

In practice, the corresponding **ground-truth action** is also projected onto the plot using the same PCA transformation. This enables a direct comparison between the modes of the policy's learned distribution and the expert's intended action, allowing for a quantitative and qualitative assessment of the policy's behavioral patterns and deviations.

## 3.2 THE INFLUENCE OF MODEL'S PERCEPTUAL CAPABILITIES

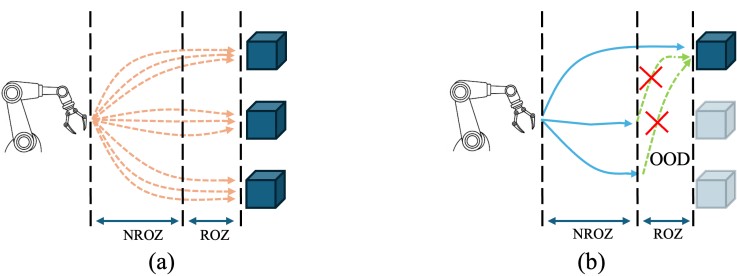

(a)                                      (b)

Figure 3: **Without considering the model's perceptual capabilities.** A mismatch between the executed trajectory and the demonstrated trajectory in traditional methods leads to task failure.

Traditional data collection often overlooks a model's perceptual capabilities, which is why even seemingly simple pick-and-place tasks struggle to achieve a high success rate. We'll use the simple task of "grasping cube at different locations" in Figure 3 to illustrate the problems with traditional data collection methods. Figure 3(a) shows a traditional data collection trajectory (orange dashed line). By training the model, we aim for the robot arm to be able to grasp cubes at different locations by following the demonstrated trajectory. However, a failure scenario like the one in Figure 3(b) may occur in actual deployment. When a cube is in the topmost position, the robot arm's trajectory (blue dashed line) might move towards the middle or lower section. Since there's a lack of corresponding demonstration data in this middle-lower region (as shown by the green dashed line), the robot arm enters an Out-of-Distribution (OOD) state, which leads to task failure. This is because a model's perceptual capabilities are limited. It simplifies complex states, such as "cubes at different locations," into "a cube in the distance," which leads to the random generation of actions. Although continuously collecting a large amount of data can sometimes accidentally supplement the states the model needs (green dashed line), this method is extremely inefficient and lacks controllability.

## 3.3 PIP-GUIDED TRAJECTORY OPTIMIZATION

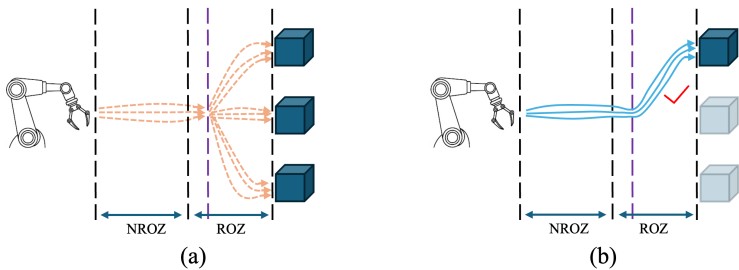

(a)                                      (b)

Figure 4: **Considering the model's perceptual capabilities.** (a) Optimized trajectory, (b) Optimized deployment results.

Using the results from PIP analysis, we can optimize data collection trajectories to mitigate the negative effects of limited perceptual capabilities. We first train a proxy model (with the same architecture as the final deployment model) using a small amount of complete demonstration data. Then, we feed the states of the complete demonstration data to the proxy model chronologically, obtaining a multimodal action distribution for each time state. This is then analyzed using Kernel Density Estimation (KDE). When the KDE results are like Figure 2(a) or 2(b), the multimodal action distribution is highly concentrated, indicating that the model can accurately predict one or more viable action plans based on the observed state. We define the region composed of these states as the Robust Operation Zone (ROZ). When the KDE results are like Figure 2(c), the multimodal action distribution is scattered and far from the ground truth. This indicates that the model cannot correctly predict the action sequence based on the current state. We define the region composed of these states as the Non-Robust Operation Zone (NROZ). As shown in Figure 4(a), we can regularize

the data trajectory in the NROZ. When the model is unable to perform the correct action based on the current state, it first moves closer to the target and then executes a precise operation upon entering the ROZ. This optimizes the data trajectory, enabling the actual deployment, as seen in Figure 4(b), to perform actions that meet human expectations, thereby ensuring task success.

## 3.4 PIP-AIDED TASK DECOMPOSITION

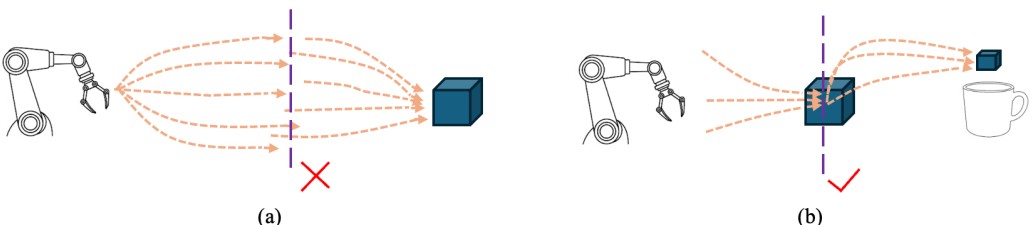

(a)                                                                 (b)

Figure 5: **Illustration of State Space Continuity in Task Segmentation.** (a) The task is segmented at a sparse location on the state trajectory. (b) The task is segmented at a concentrated location on the state trajectory. The purple dashed line represents the task segmentation point.

Most dominant robot imitation learning models are based on the Markov Decision Process (MDP) Bellman (1957), which means they rely solely on the current state to predict actions. Therefore, we can break a complete task down into a series of subtasks for data collection. However, arbitrary task decomposition can lead to issues such as disjointed subtasks and discontinuous actions. For effective subtask segmentation, two core requirements must be met: **A) Ensure state space continuity:** Hamilton (1994). As shown in Figure 5(a), cutting at a location with a large state space range can make it difficult for subtask states to connect, potentially causing action pauses. Conversely, as shown in Figure 5(b), cutting at a location with a smaller state space range is easier to connect, ensuring smooth and continuous actions.

**B) Decompose subtasks within the ROZ:** The endpoint of one subtask is typically the starting point of another. This requires the policy to be able to switch from the old policy $p_{\theta,old}(A|s, L_{old})$ to the new policy $p_{\theta,new}(A|s, L_{new})$ at a specific transition state $s_{trans}$. To determine if $s_{trans}$ is suitable, we can calculate the **Shannon Entropy** Lin (2002) of the policy's action distribution:

$$H(A|s, L) = -\sum_A p_\theta(A|s, L) \log p_\theta(A|s, L)$$

If the $s_{trans}$ is **located within the NROZ**, the action distribution entropy of the old policy is very high ($H(A|s_{trans}, L_{old}) \gg 0$). This means that when the model attempts to complete the final step of the old task, its actions are random and unpredictable. This uncertainty prevents it from reliably executing the correct transition action, making it unable to smoothly enter the new task's starting state and causing the policy switching to fail. Conversely, if $s_{trans}$ is **located within the ROZ**, the old policy's action distribution entropy is extremely low ($H(A|s_{trans}, L_{old}) \approx 0$), which ensures that the transition from the old state to the new state is stable and predictable. The model can accurately perceive this clear state change, which reliably triggers the call to the new policy.

## 3.5 MODEL-IN-THE-LOOP DATA OPTIMIZATION FRAMEWORK(MIL-DOF)

We propose a Model-in-the-Loop data collection method. This approach uses feedback from a proxy model to optimize data collection trajectories. By training the model with both supplementary subtask data and the original task data, we can improve its performance. This forms an efficient data flywheel, which significantly boosts data collection efficiency and task success rates.

### 3.5.1 SUBTASK-LEVEL DATA AUGMENTATION

Combining complete task data with subtask data for training is a highly efficient data collection method that also boosts model performance. This approach, which includes a subtask-level data augmentation strategy, offers the following key advantages:

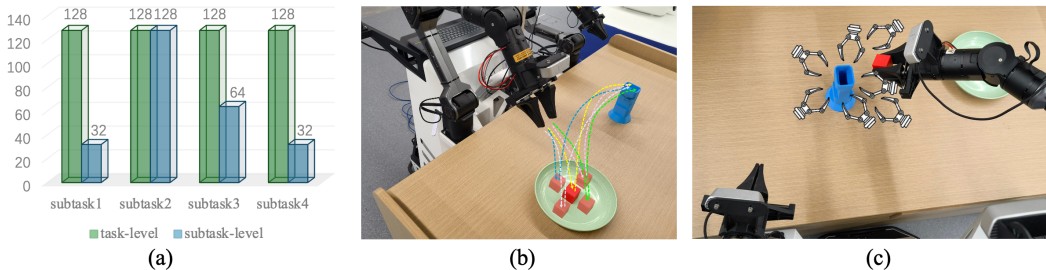

(a)                (b)                (c)

Figure 6: **The advantages of subtask-level data augmentation.** (a) Reduces overall data volume compared to task-level approaches. (b) Provides targeted data augmentation based on task difficulty. (c) Supplements state space at precise manipulation locations.

**Reduced Overall Data Volume:** By decomposing complex, long-horizon tasks into multiple sub-tasks, we can perform targeted data collection based on each subtask's complexity. This approach eliminates the need to repeatedly collect data for the entire task due to a single complex stage, significantly reducing the total data volume required compared to traditional methods.

**Targeted Augmentation for Complex Subtasks:** Some subtasks, such as those requiring precise positioning or positional generalization, are inherently more challenging and demand a greater diversity of data. We can augment the model's capability for performing complex subtasks.

**Proactive Failure Recovery:** Due to the accumulation of errors, a robot arm may enter Out-of-Distribution (OOD) regions during task execution. Our approach supports the proactive augmentation of data in these specific areas, enabling the model to recover from potential failures and substantially increasing system reliability.

### 3.5.2 SUBTASK-LEVEL DATA CLEANING

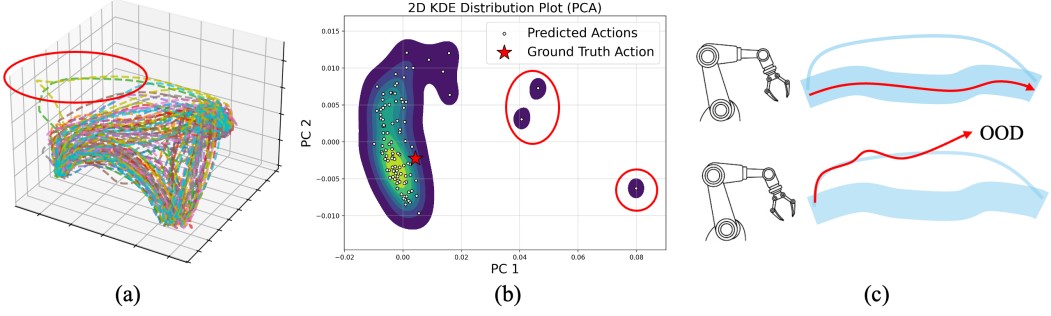

(a)                (b)                (c)

Figure 7: **The Impact of Anomalous Trajectories.** (a) Abnormal trajectory (within the red circle) in real-world task data. (b) The action distribution of anomalous trajectories in a Kernel Density Estimation (KDE) plot. (c) Impact of an Abnormal Trajectory on Real-World Deployment.

As shown in Figure 7(a), the trajectory within the red circle significantly deviates from the normal path. This type of data, though technically correct, negatively impacts model performance. Through Policy-Intent Probe (PIP) analysis, we found that the Kernel Density Estimation (KDE) results in Figure 7(b) show a non-zero probability of the model generating such significantly deviated action sequences. In actual deployment, as shown in Figure 7(c), the cumulative effect of errors can cause the model to exceed the data collection range (the blue area) and enter an Out-of-Distribution (OOD) state, ultimately leading to task failure. Therefore, both this type of anomalous data and data that failed during collection must be cleaned. Our subtask-level data cleaning method efficiently solves this problem: we only need to remove the subtask containing the anomalous or failed trajectory, without having to discard the entire full task data. This approach not only ensures data quality but also significantly improves the utilization of valid data.

## 4 EXPERIMENTS

We design experiments to answer the following questions: **Q1:** Will the perception gap between humans and robots affect task success rate? **Q2:** What is the difference between the PIP-based segmentation strategy and direct subtask segmentation? **Q3:** Does our data optimization framework improve policy performance? **Q4:** Can our method improve data collection efficiency?

### 4.1 SETUP

**Platform**   All experiments in this study were conducted using the same robotic platform: the AgileX Cobot Magic with Piper, which shares the same system as Mobile ALOHA Fu et al. (2024). We collected data using the same leader-follower teleoperation setup as ALOHA Zhao et al. (2023b). The dataset, intended for model training and inference, comprises the joint angles of the two follower arms and three cameras: one front camera for a global view and two wrist cameras for local views.

**Baselines**   We use two flow-based vision-language-action (VLA) models as our baselines: $\pi_0$ Black et al. (2024) and the more advanced $\pi_{0.5}$ Intelligence et al. (2025) which offers better open-world generalization. Because they represent mainstream VLA models with good performance on real-world tasks. Our focus is on data, and by using these models, we can avoid performance issues caused by model-specific factors. We uniformly train all tasks for 30,000 steps, using the original parameters without any modifications.

**Task**   To evaluate our optimization framework, we used three real-world tasks, including a precise manipulation task and two long-horizon manipulation tasks. **A) Precision pick-and-place with positional generalization:** As shown in Figure 5(b), this task requires precisely placing the red block from a plate into a blue cup. The block's position on the plate is random. **B) Retrieve item from drawer:** The robotic arm needs to pull open a closed drawer, take out a teddy bear, place it in the basket in front, and then close the drawer. The initial position of the teddy bear inside the drawer is random. **C) Rice Cooker operation:** Requires the robotic arm to open the rice cooker by pressing a button, pour rice from a cup into it, and finally close the rice cooker. The position of the cup on the right side of the table is random. For each task, we established scoring criteria based on the task procedure and measured the task execution duration as well as data collection efficiency. Details of the evaluation and the flow diagrams for the three tasks can be found in Appendix C.

### 4.2 MAIN RESULTS

To answer the questions raised in Section 4, we conducted a series of comparative experiments on Task A, which requires high-precision manipulation, under the condition that the total data frames remain nearly consistent. Our approach involved four primary data collection planes. **Plane A:** Full task data collected without considering the model's perception capabilities. **Plane B:** Full data and subtask data manually segmented by a person. **Plane C:** Full task data collected by considering the model's perception capabilities. **Plane D:** Our proposed model-in-the-loop data optimization method. Based on the results in Table 1, the following observations can be made:

Table 1: Comparing four data collection Plane on Task A.

|  | Metric | Plane A | Plane B | Plane C | Plane D |
|---|---|---|---|---|---|
| Task A | Success Score ↑ | 55% | 35% | 73% | **81%** |
|  | Horizon Score ↑ | 81% | 20% | 85% | **88%** |

1. **Manual subtask segmentation might degrade model performance:** During real-world deployment, Plane B often caused the robot arm to halt during subtask transitions or resulted in task failures due to choppy execution, leading to a noticeably low **horizon score**.

2. **Considering the model's perception capabilities can improve performance:** Plane A's failures were mainly due to poor object position estimation, often resulting in inaccurate

grasps when picking up cubes. In contrast, Plane C was able to locate cubes more accurately, achieving precise grasps.

3. **The model-in-the-loop plane performed best:** Building on Plane C, Plane D effectively incorporated additional subtask data for grasping cubes at various locations. Furthermore, it demonstrated a certain degree of failure recovery, placing cubes into the cup more accurately and without the choppy execution or poor task transitions seen in Plane B, thus achieving **state-of-the-art** performance among the four Planes.

To demonstrate that our optimization framework improves model performance, we conducted a comparison of traditional data collection methods with our proposed method on three real-world tasks. For all tasks, a consistent data collection duration of 100 episodes was used. Based on the results in Table 2, the following observations can be made:

Table 2: Experimental Results on Three Tasks. S-Score: Success Score, H-score: Horizon Score.

| | | Task A | | Task B | | Task C | |
|---|---|---|---|---|---|---|---|
| | | S-Score ↑ | H-Score ↑ | S-Score ↑ | H-Score ↑ | S-Score ↑ | H-Score ↑ |
| $\pi_0$ | w/o Ours | 65% | 86% | 85% | 84% | 23% | 8% |
| | w/ Ours | 85% | **88**% | 92% | 87% | 62% | 50% |
| $\pi_{0.5}$ | w/o Ours | 69% | 85% | 87% | 82% | 26% | 12% |
| | w/ Ours | **86**% | 84% | **95**% | **92**% | **63**% | **55**% |

1. **Significant Performance Improvement:** Compared to traditional data collection methods, our optimization framework demonstrates a significant performance improvement and achieving **state-of-the-art** performance among the three tasks. Specifically, our method allows the robotic arm to grasp cubes more precisely in Task A, and in Task C, it enables a more accurate opening of the rice cooker lid, leading to smoother task execution.

2. **Higher Data Collection Efficiency:** Traditional methods of stacking data to improve model performance are extremely inefficient. As seen in Table 2, the amount of data for Task A is far greater than in Table 1, yet the increase in success rate is minimal, falling short of the performance achieved by our method in Table 1. This fully demonstrates that our method has much higher data collection efficiency.

3. **Adaptability to High-Precision Tasks:** Our method is better suited for tasks that require high-precision operations. For Task B, although its process is longer but relatively simple, the performance improvement is comparable to traditional collection methods. However, for Tasks A and C, the performance gains are much more significant.

## 5 CONCLUSION

This paper presents a novel model-in-the-loop data optimization framework, aimed at resolving task failures in robotic imitation learning caused by a mismatch between the policy and human expectations. The core of this framework lies in using a Policy Intent Probe (PIP) to gain deep insights into a model's perceptual capabilities by analyzing the policy distribution of a proxy model. Based on this feedback, we successfully bridged the human-robot perceptual gap and significantly improved task success rates. To further enhance efficiency and robustness, we use PIP to assist with subtask decomposition, ensuring fluid actions and smooth transitions between subtasks. Simultaneously, subtask-level data augmentation effectively boosts data efficiency and model robustness, while subtask-level data cleaning increases data utilization. Our framework directly integrates the model's capabilities into the data collection process, forming an efficient data flywheel. This not only breaks the vicious cycle of blindly collecting data but also, by comprehensively improving data quality and efficiency, significantly enhances the success rate and robustness of complex long-horizon manipulation tasks, achieving **state-of-the-art** performance on three real-world tasks.

## 6 REPRODUCIBILITY STATEMENT

All our data collection and experiments were conducted in the real world. We saved all the experimental model training checkpoints, which allows for the reproduction of the experimental process by recreating the experimental environment. Although minor variations in lighting and other environmental factors may lead to slight changes in the results, the overall experimental conclusion remains consistent.

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

## A ETHICS STATEMENT

The authors of this paper strictly adhere to the ICLR Code of Ethics. The researchers have not knowingly made false or misleading claims, fabricated or manipulated data, or misrepresented research findings. We received assistance from researchers outside of the author group for collecting the real-world data used in this paper. Due to the limited scope of their contribution, they were not included as authors, which presents a potential conflict of interest. The hardware used for the experiments in this paper is part of a shared resource, and its use may have temporarily impacted the work of other researchers, potentially affecting their studies.

## B THE USE OF LARGE LANGUAGE MODELS

To enhance the clarity and professionalism of this paper, we utilized Gemini 2.5 Pro and Gemini 2.5 Flash for language refinement. For the initial draft, we incorporated structural suggestions provided by Large Language Models (LLMs). Additionally, we used Claude Sonnet 4 to assist with code debugging.

## C  TASK EVALUATION METRIC

### C.1  HORIZON SCORE

The average time taken to successfully complete the task, which measures the policy's efficiency.

$$Horizon_{score} = \left(\frac{1}{n}\sum_{i=1}^{n}\left[1 - \min\left(1, \frac{|x_i - \mu|}{\mu}\right)\right]\right) \times 100\%$$

### C.2  SUCCESS SCORE

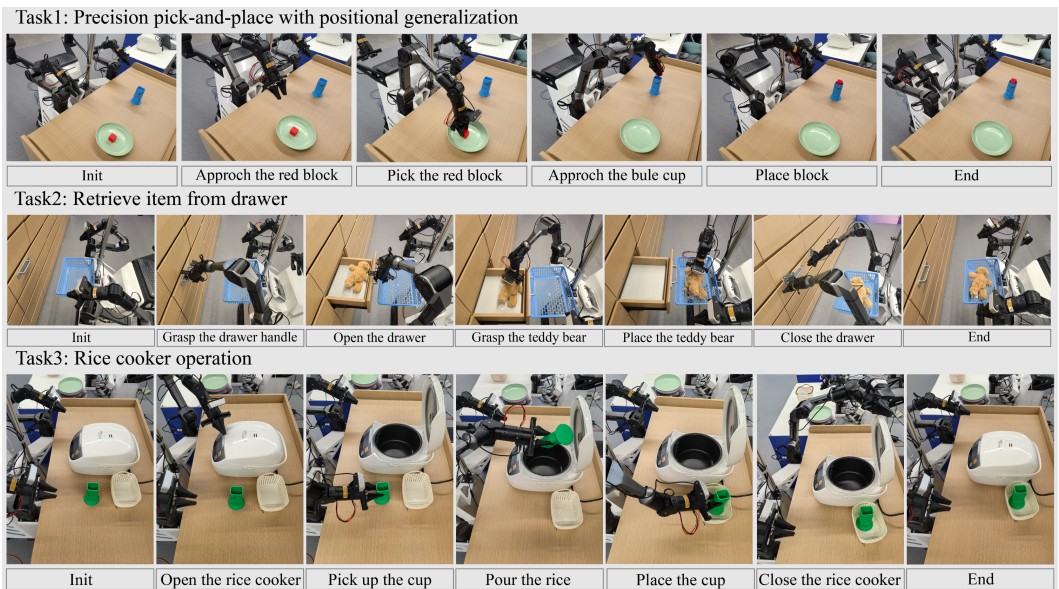

Figure 8: Flow diagrams for three Real-World tasks

We cover three tasks: Precision pick-and-place with positional generalization, Retrieve item from drawer, and Rice cooker operation, and have established scoring criteria for them. During the experiment, each task is executed 50 times. The scores for all steps are tallied to determine the final task score, with a maximum score of 1 and a minimum score of 0. Figure 8 shows the flow diagrams for the different tasks. The scoring criteria for the different tasks are as follows:

1) Precision pick-and-place with positional generalization

**Task description.** This task demands the precise manipulation of the robotic arm and requires the generalization ability of the cup's position. The task consists of four steps: first, the right arm approach the red block; second, the right arm picks up the red block; third, the right arm places the red block into the blue cup; and fourth, the right arm returns to the original position. The scoring criteria are as follows:

· **Step 1: Right arm approaching the red block**

  - **scoring 0:** The gripper does not approach the red block.
  - **scoring 0.5:** The gripper approaches the red block, but is either too far or too close.
  - **scoring 1:** The gripper approaches the red block at the correct position.

· **Step 2: Right arm picking up the red block**

  - **scoring 0:** The gripper does not grasp the red block.
  - **scoring 0.5:** The gripper grasps the red block, struggles to maintain a stable grasp, but finally succeeds.

- **scoring 1:** The gripper grasps the red block successfully without any slippage.

· **Step 3: Right arm placing the red block**

- **scoring 0:** The gripper does not place the red block into the blue cup.
- **scoring 0.5:** The gripper places the red block into the blue cup, but significantly moves or contacts the cup.
- **scoring 1:** The gripper successfully places the red block into the blue cup without any issues.

· **Step 4: Right arm returning to the original position**

- **scoring 0:** The gripper does not return to the original position.
- **scoring 0.5:** The gripper returns to the original position, but is far from the correct position.
- **scoring 1:** The gripper successfully returns to the original position.

2) Retrieve item from drawer

**Task description.** This task is a complex, bimanual manipulation task, which requires robot using both arms to retrieve the teddy bear from the drawer accurately. This task consists of five steps: first, the left arm opens the drawer; second, the right arm grasps the neck of the teddy bear; third, the right arm places the teddy bear in the basket in front; and fourth, both arms close the drawer. The scoring criteria are as follows:

· **Step 1: Left arm opening the drawer**

- **scoring 0:** The left arm does not open the drawer.
- **scoring 0.5:** The left arm grasps the handle of the drawer and opens it slightly.
- **scoring 1:** The left arm successfully opens the drawer completely.

· **Step 2: Right arm grasping the teddy bear**

- **scoring 0:** The gripper does not grasp the teddy bear.
- **scoring 0.5:** The gripper grasps the teddy bear, struggles to maintain a stable grasp, but finally succeeds.
- **scoring 1:** The gripper successfully grasps the teddy bear without any slippage.

· **Step 3: Right arm placing the teddy bear**

- **scoring 0:** The right arm does not place the teddy bear in the basket in front.
- **scoring 0.5:** The right arm places the teddy bear in the basket, but significantly moves or contacts the drawer or basket.
- **scoring 1:** The right arm places the teddy bear in the basket without any issues.

· **Step 4: Both arms closing the drawer**

- **scoring 0:** Both arms do not close the drawer.
- **scoring 0.5:** Both arms push the drawer and close it slightly.
- **scoring 1:** Both arms successfully close the drawer completely.

3) Rice cooker operation

**Task description.** This task is a long-term sequence operation problem, which requires the robotic arm to perform a series of coordinated actions to operate the rice cooker. The task consists of five steps: first, the left arm presses the button of the rice cooker and opens it; second, the right arm grasps the cup on the desk; third, the right arm pours the rice from the cup into the rice cooker; fourth, the right arm places the cup in the basket; and fifth, the left arm closes the rice cooker. The scoring criteria are as follows:

· **Step 1: Left arm opening the rice cooker**

- **scoring 0:** The left arm does not open the rice cooker.
- **scoring 0.5:** The left arm presses the button and opens the cooker, but significantly moves or contacts the rice cooker.
- **scoring 1:** The left arm successfully opens the rice cooker without any issues.

· **Step 2: Right arm grasping the cup**

- **scoring 0:** The right arm does not grasp the cup.
- **scoring 0.5:** The right arm grasps the cup,struggles to maintain a stable grasp, but finally succeeds.
- **scoring 1:** The right arm successfully grasps the teddy bear without any slippage.

· **Step 3: Right arm pouring the rice**

- **scoring 0:** The right arm does not pour the rice.
- **scoring 0.5:** The right arm pours the rice, but significantly moves or contacts the rice cooker.
- **scoring 1:** The right arm successfully pours the rice without any issues.

· **Step 4: Right arm placing the cup**

- **scoring 0:** The right arm does not place the cup in the basket.
- **scoring 0.5:** The right arm places the cup, but significantly moves or contacts the basket.
- **scoring 1:** The right arm successfully places the cup without any issues.

· **Step 5: Left arm closing the rice cooker**

- **scoring 0:** The left arm does not close the rice cooker.
- **scoring 0.5:** The left arm closes the rice cooker but significantly moves the rice cooker.
- **scoring 1:** The left arm successfully closes the rice cooker without any issues.

