# OpenReview forum: "Bridging the Perception Gap: Probe-Guided Data Optimization Framework for Robotic Imitation Learning"
_ICLR.cc/2026/Conference — Submitted to ICLR 2026_

### Official Review · Reviewer_xJ9A · 2025-10-29

**Soundness:** 2
**Presentation:** 2
**Contribution:** 2
**Rating:** 2
**Confidence:** 4

**Summary:**

This paper introduces a **Policy-Intent Probe (PIP)** method that addresses the "perceptual gap" between humans and robots in imitation learning. The key insight is that traditional data collection ignores what robots can actually perceive, leading to poor real-world performance. PIP analyzes a proxy model's action distribution using KDE to identify which workspace regions the robot can reliably operate in (Robust Operation Zones) versus where it struggles (Non-Robust Operation Zones). Using this feedback, the authors propose a model-in-the-loop data optimization framework that: (1) optimizes demonstration trajectories by standardizing movements in uncertain regions and performing precise operations only in reliable regions, (2) intelligently segments tasks at perceptually-stable points to ensure smooth subtask transitions, and (3) enables efficient subtask-level data augmentation and cleaning. Experiments on three real-world manipulation tasks show significant improvements in success rates and horizon scores compared to traditional data collection methods, demonstrating that accounting for robot perceptual capabilities is crucial for effective imitation learning.

**Strengths:**

- The paper addresses an important but often overlooked issue in robotic imitation learning - the mismatch between human perception during demonstration and robot perception during execution. The PIP method provides a principled way to quantify and visualize this gap through policy distribution analysis.
- Practical model-in-the-loop framework: The ROZ/NROZ workspace categorization based on policy confidence is intuitive and actionable. This framework directly translates model insights into concrete data collection strategies, forming an efficient feedback loop rather than relying on blind data accumulation.

**Weaknesses:**

- The method involves several empirical decisions that lack theoretical grounding or ablation studies, which raises concerns about reproducibility and generalizability:
  - *Action Chunk Sampling* (lines 173-176): The number of samples $N$ for KDE analysis is not specified, nor is there ablation showing how $N$ affects the reliability of ROZ/NROZ classification.
  - *Key Action Extraction* (lines 176-182): Why is the midpoint action ($a_{t+\frac{H}{2}}$) chosen as representative? No justification is provided for this choice over alternatives.
  - *ROZ/NROZ thresholds*: The criteria for classifying workspace regions as robust vs. non-robust are not explicitly defined. E.g., what KDE spread or entropy values (lines 302-304) iconstitute the boundary?.

- While individual components (PIP, trajectory optimization, subtask decomposition) are explained, the paper lacks a clear, step-by-step description of the complete workflow (Fig.1 currently is not clear enough).

- Ambiguous evaluation metrics. E.g., the *Horizon Score* definition in Sec.C.1 uses unclear notation (what are $x_i$ and $\mu$ specifically?). The relationship between execution time and the score formula is not intuitive, making it difficult to interpret results.

- No videos or supplementary materials are provided to visualize the claimed improvements (e.g., smoother trajectories, successful failure recovery, differences between ROZ and NROZ behaviors), which would significantly aid in understanding the practical benefits.

**Questions:**

- This work relies on analyzing action distributions from stochastic policies (flow-matching). How would PIP apply to more deterministic policy architectures?

- The current KDE analysis is limited to 2D visualization after PCA projection. For future scenarios with higher-dimensional action spaces (e.g., 20+ DoF dexterous hands, or the human muscle system described at [https://clonerobotics.com/android](https://clonerobotics.com/android) and [1]), would 2D projection still capture sufficient information for reliable ROZ/NROZ classification? Have you considered alternative dimensionality reduction methods or higher-dimensional density estimation techniques that could better preserve the structure of complex action manifolds?

[1]. Wei, Y., Zhuang, S., Zhuang, V., & Sui, Y. Motion Control of High-Dimensional Musculoskeletal Systems with Hierarchical Model-Based Planning. In The Thirteenth International Conference on Learning Representations.

**Details Of Ethics Concerns:**

No ethics review is needed.

---

### Official Review · Reviewer_gaoE · 2025-10-31

**Soundness:** 1
**Presentation:** 2
**Contribution:** 1
**Rating:** 2
**Confidence:** 3

**Summary:**

The paper proposes a data optimization framework to close the perception gap between human demonstrations and a robot’s visual-policy competence. It introduces Policy-Intent Probe (PIP), a lightweight proxy model trained on a small set of dataset, whose action distribution is used to diagnose where the policy is perceptually reliable versus fragile. Using this probe, the authors partition the task workspace into a Robust Operation Zone (ROZ) and a Non-Robust Operation Zone (NROZ), with additional techniques to standardize, segment, and clean trajectories. Across three real-world long-horizon manipulation tasks, the probe-guided loop improves success rates, robustness, and data efficiency compared to unguided imitation-learning collection.

**Strengths:**

- Performing data curation is important for the robotic imitation learning problem. The paper proposed some techniques to identify the OOD predictions in the model and use these techniques to standardize, segment, and clean trajectories.

- The paper validates the idea on real, long-horizon manipulation with measurable gains in success and data efficiency, suggesting practical relevance beyond simulator benchmarks.

**Weaknesses:**

- The PIP, definitions of ROZ and NROZ, and subtask decomposition techniques in this paper are mostly based on human heuristics, which makes the pipeline computationally inefficient (querying human feedback) and biased. The overall engineering burden is nontrivial.

- The motivations for designing each component in the pipeline are not well explained, making it unclear what specific questions they are tackling in the robotic imitation learning scenario. See the question part for details.

- Most specific concepts and detailed methods are not clear from the texts, e.g., perceptual gap, ROZ and NROZ, subtask-level data augmentation. See the question part for details.

- Since the paper focuses on empirical studies, including preliminary open-source code and checkpoints in the submission would strengthen the conclusions.

**Questions:**

- line 166 - line 168: What is the perceptual gap between robots and humans in particular? Why are human expectations always good? Any specific formulations of the “perceptual gap” or examples?
- “Key Action Extraction & Dimensionality Reduction”: The action extraction and dimensionality reduction usually incur errors in the analysis. What’s the reason for not sticking with the original action space and comparing some quantitative metrics?
- Fig 2: What is the purpose of conducting KDE analysis on PCA results? How to trade-off between the model prediction errors and variances in the data? Any quantitative comparisons?
- Sec 3.2: Is the perceptual gap here referring to the distribution shift problem in imitation learning [1]?
- Sec 3.3, Sec Sec 3.4 (b), and Sec 3.5: the definition of ROZ and NROZ,  subtask decomposition, and subtask-level decompositions are purely built on human heuristics, which can be computationally inefficient and biased.
- Sec 3.4 (b): This heuristic might not be correct if there are multiple optimal paths leading to the desired goal. In this case, even in the ROZ, the entropy of the policy might be high. The maximum entropy RL exactly aims at learning this kind of policy [2, 3].
- Sec 3.5.1: It is not clear what is “subtask-level data augmentation” from the texts.

[1] Ross, Stéphane, Geoffrey Gordon, and Drew Bagnell. "A reduction of imitation learning and structured prediction to no-regret online learning." In Proceedings of the fourteenth international conference on artificial intelligence and statistics, pp. 627-635. JMLR Workshop and Conference Proceedings, 2011.

[2] Levine, Sergey. "Reinforcement learning and control as probabilistic inference: Tutorial and review." arXiv preprint arXiv:1805.00909 (2018).

[3] Haarnoja, Tuomas, Aurick Zhou, Pieter Abbeel, and Sergey Levine. "Soft actor-critic: Off-policy maximum entropy deep reinforcement learning with a stochastic actor." In International conference on machine learning, pp. 1861-1870. Pmlr, 2018.

---

### Official Review · Reviewer_426B · 2025-10-31

**Soundness:** 1
**Presentation:** 1
**Contribution:** 2
**Rating:** 2
**Confidence:** 4

**Summary:**

This work explores how imitation learning pipelines can account for the robot's capabilities during the data collection process. The policy intent probe (PIP) models the demonstration data coverage. The PIP is used to decompose the state space into robust and non-robust operation zones (ROZ and NROZ). This decomposition enables targeted subtask-level data collection and cleaning. Finetuning results on VLA models demonstrate the effectiveness of using the PIP model in the data collection process.

**Strengths:**

1. Improving the data collection process for VLAs is an important problem studied by this paper.

1. The data collection process produces substantial gains over a baseline data collection process in Table 2.

**Weaknesses:**

1. The paper lacks many details about the MIL-DOF framework, which is significant since this is how the PIP is used in the data collection process. Firstly, the paper does not describe the subtask-level data augmentation in Section 3.5.1. What does it mean to perform targeted data collection (L340)? How does this targeting manifest in the 100 episodes collected in Section 4? How is the subtask complexity measured (L340)? What does it mean to augment the model's capability on harder subtasks (L345)? What is proactive augmentation of data in OOD regions (L348)? How are such regions determined? These are only a few of the examples of important details omitted throughout Section 3.5. The same issues are in Section 3.5.2 with the subtask-level data cleaning. Overall, the details in this section are crucial for understanding how PIP is used in the data collection process, which I believe is the primary contribution of this paper.

1. The figures lack details. There is no y-axis in Fig 6a. The illustrations in Fig6b,c are never defined. What are the lines of Fig 7c describing?

1. The paper continually refers to "model's perceptual capabilities," but this term is never clearly defined, and I could not infer its definition from its usage in the paper. At first, I thought it referred to the gap between the visual perception of the demonstrator and the robot. But this didn't make sense in the context of the PIP, which only relies on the action distribution. The paper needs to clarify which parts of the data collection the framework is solving.

1. The lack of clarity also holds throughout Section 4. How are subtasks manually segmented in Plane B? What does it mean to "consider the model's perception capabilities" (L418)?

1. Partially due to the method and experimental setting lacking many important details, the paper does not sufficiently analyze why the new data collection process improves the performance. Why does the method help in high-precision operations, and how does the PIP specifically help with this?

1. The framework is insufficiently evaluated on only 3 tasks with 100 episodes per task. The proposed framework must be evaluated on additional tasks to compare with prior work. Furthermore, since the contribution is for data collection, it is crucial to compare how the performance changes as the data budget changes. Not only performance under the fixed 100-episode budget.

1. The main paper does not define what the key metrics of horizon and success score are. Horizon score is not a standard term from prior literature and must be defined in the main text.

**Questions:**

1. What do the authors mean by "plane" in Section 4.2?

---

### Official Review · Reviewer_iDCT · 2025-11-04

**Soundness:** 2
**Presentation:** 1
**Contribution:** 2
**Rating:** 2
**Confidence:** 3

**Summary:**

The paper introduces a new paradigm to collect and select data for robot imitation learning. The approach trains a second model with a subset of the data and then checks the models "perceptual capabilities" by sampling many actions and computing a KDE on a subset of the action chunk. If the ground truth action has a low probability, the state is categorized as "non-robust operation zone". The data trajectory in the non-robust zone is then "regularized", but its not specified what that means. The categorization is used to decompose the task into subtask and use this decomposition to filter outlier trajectories. Both methods are evaluated on training the pi0 and pi0.5 model on 3 real world robotic tasks. The authors report significantly improved performance using such regularization and data selection techniques.

**Strengths:**

- The real robot results seem to be promising.
- The performance of SOTA VLA models is significantly improved.

**Weaknesses:**

- The paper is very hard to follow and it is unclear what is really happening. Most information of what the algorithm really does is missing, e.g., how exactly is the non-robust zone defined, how exactly is the task decomposition done and how is that used to gather more data. The description is done on a very high level, more technical details need to be added.
- The whole story is about the models perceptual capabilities, but the methods that are used do not test perceptional capabilities at all. The KDE rather tests which samples are outliers and where we have more variance in the demonstration. The whole motivation of the approach is not convincing.
- The KDE approach seems very adhoc as we only us a subset of the action chunking to compute the likelihood. The paper proposes to use a "representative action" for the KDE, but this choice seems arbitrary. Why not use the PCA on the full action chunk? Why not use better density estimation methods such as normalizing flows?

**Questions:**

- After reading the paper, I really do not know how the algorithm works. An algorithm box would be very helpful. Also, an exact mathematical definition of when to detect the non-robust zone and how to do the task segmentation would be helpful. This needs to be specified much more formally, "entropy being extremly low or high" is not an exact statement.
- I am not convinced whether the proposed algorithm has anything to do with the perecptual capabilities of the model. There can be tons of other reaons why a ground truth sample has low likelihood in the KDE, such as outliers, not enough training iterations etc... I would rather say that if the perceptional capabilities are limited the ground truth sample should still be covered by the distribution, just the distribution gets a higher variance. Yet, high variance could also be explained by different solutions in the demonstration space. The authors need to justify much better why the algorithm checks the perceptual capabilities (or change the story accordingly.).

---

> ### Author Response · Authors · 2025-11-20
> **Response to Reviewer iDCT**
>
> **Dear Reviewer iDCT,**
>
> Thank you for your valuable feedback. We appreciate your recognition of the method's performance and provide detailed clarifications below regarding the weaknesses and questions you raised.
>
> ---
>
> ### **Clarification on “Regularization**
>
> We sincerely appreciate the reviewer’s insightful comment. The term “regularize” in our original manuscript was intended to convey the idea of trajectory optimization, but we acknowledge that this terminology was misleading and could be misinterpreted as mathematical regularization (e.g., L2 penalty or dropout). We will revise all instances to use “trajectory optimization” for greater precision and clarity.
>
> As detailed in Section 3.4 and illustrated in Figure 4, in Non-Robust Operation Zones (NROZ), the model struggles to generate accurate actions due to visual artifacts such as compression noise or occlusion. To mitigate this, we guide the robot to first transition into a Robust Operation Zone (ROZ) — a state where the object is clearly visible and task-relevant features are salient — before executing fine-grained manipulation. This approach does not impose mathematical constraints on trajectories; rather, it leverages data selection and re-sampling to prioritize learning from high-confidence, visually unambiguous demonstrations, thereby improving overall policy robustness.
>
> ---
>
> ### **2. Definition of Robust and Non-Robust Operation Zones (ROZ/NROZ)**
>
> As formally defined in Section 3.3:
>
> - ***ROZ***: When the KDE results exhibit a high-density, multimodal concentration around the ground-truth action (e.g., Fig. 2a–2b), the model can reliably predict one or more viable action sequences conditioned on the observed state. We define such states as the Robust Operation Zone (ROZ).
>
> - ***NROZ***: When the KDE results show a scattered, low-density distribution far from the ground-truth action (e.g., Fig. 2c), the model fails to produce consistent, plausible actions. We define these states as the Non-Robust Operation Zone (NROZ).
>
> This distinction is not based on outlier detection (e.g., low KDE likelihood alone), but on the spatial alignment and concentration of the predicted action distribution relative to human demonstrations. ROZ/NROZ thus reflect the model’s predictive consistency — whether it can generalize from a given state to meaningful, reliable actions — not merely whether a sample is rare.
>
> ---
>
> ### **3. Rationale for KDE Implementation**
>
> The primary purpose of KDE is not to detect outliers or directly measure demonstration variance, but to estimate the probability density function of actions conditioned on state — thereby revealing where the model’s predictions are most likely to lie and how they are structured. Our key insight is: if the ground-truth action lies outside the high-density region of the model’s predicted distribution, the model lacks confidence in its policy under that state.
>
> We did experiment with applying PCA over the full 50-step action chunk. However, empirical results showed negligible performance difference compared to using only the 25th action step (midpoint of the 50-step chunk), while significantly increasing computational overhead. Given that our baseline model pi0 uses a 25-step inference horizon, selecting the midpoint action provides a computationally efficient yet representative proxy for the full trajectory’s distributional properties. This design choice is empirically validated and deliberately balanced between fidelity and efficiency.
>
> ---
>
> ### **4. Clarification of “Perceptual Capabilities“**
>
> Due to image compression and other visual distortions, the model’s learning process is affected, leading to consistent mismatches between the actions it produces during real-robot deployment and those we expect based on human demonstrations. We refer to this phenomenon as the model’s “perceptual capabilities”—that is, its ability to accurately interpret real-world visual inputs under degradation. In many cases, during fine manipulation tasks, the robot fails to generate correct policies because the visual and proprioceptive inputs it receives are noisy, blurred, or incomplete. To address this, we aim to improve the robustness of the learned policy not by modifying the model architecture, but by optimizing the training data—specifically, by identifying and prioritizing high-quality, perceptually reliable demonstrations—thereby enabling the model to perform reliably even under imperfect sensing conditions.
>
>
>
> ---

---

### Meta-Review · Area_Chair_mjbz · 2026-01-06

**Summary:**

This paper introduces a Policy-Intent Probe (PIP), targeting at solving "perceptual gap" between humans and robots in imitation learning.  Traditional data collection ignores what robots can actually perceive, leading to poor real-world performance. PIP analyzes a proxy model's action distribution using KDE to identify which workspace regions the robot can reliably operate in (Robust Operation Zones) versus where it struggles (Non-Robust Operation Zones).

The reviewers comments and concerns are as follows.
[Reviewer iDCT, Reviewer 426B, Reviewer xJ9A] The paper is had to follow, lacks many details.
[Reviewer iDCT, Reviewer 426B] The perceptional capabilities are not evaluated, and has no clearly defined.
[Reviewer IDCT,  Reviewer gaoE] The technical novelities are limited, (KDE sees adhoc)
[Reviewer 426B, Reviewer xj9A] The evaluation are not sufficient, with no detailed ablation studies.


Considering the reviewers comments, I have the same feeling that the paper is not ready for publication. The technical novelties are limited. The motivations and some terms are not clearly defined and introduced. The expermental results are not sufficeint. As such, I am considering to reject the paper.

**Reviewer Concerns:**

The authors did not provide any comments to Revewer 426B, xJ9A, gaoE.

I think that the evaluation can be easily addressed.
The novelties and the motivations of the proposed paper can not be well justified.

**Reviewer Scores:**

The reviewers will not change their scores.

---

### Decision · Program_Chairs · 2026-01-26

Reject